# The Problem of Meaning in AI and Robotics: Still with Us after All These Years

**Tom Froese** [1,2,*] **and Shigeru Taguchi** [3]

[1]   Institute of Applied Mathematics and Systems Research (IIMAS), National Autonomous University of Mexico (UNAM), Mexico City 04510, Mexico
[2]   Center for the Sciences of Complexity (C3), National Autonomous University of Mexico (UNAM), Mexico City 04510, Mexico
[3]   Faculty of Humanities and Human Sciences, Hokkaido University, Sapporo 060-0810, Japan; tag@let.hokudai.ac.jp
*   Correspondence: t.froese@gmail.com

**Abstract:** In this essay we critically evaluate the progress that has been made in solving the problem of meaning in artificial intelligence (AI) and robotics. We remain skeptical about solutions based on deep neural networks and cognitive robotics, which in our opinion do not fundamentally address the problem. We agree with the enactive approach to cognitive science that things appear as intrinsically meaningful for living beings because of their precarious existence as adaptive autopoietic individuals. But this approach inherits the problem of failing to account for how meaning as such could make a difference for an agent's behavior. In a nutshell, if life and mind are identified with physically deterministic phenomena, then there is no conceptual room for meaning to play a role in its own right. We argue that this impotence of meaning can be addressed by revising the concept of nature such that the macroscopic scale of the living can be characterized by physical indeterminacy. We consider the implications of this revision of the mind-body relationship for synthetic approaches.

**Keywords:** mind-body problem; 4E cognition; cognitive robotics; artificial life; minimal cognition; dynamical approach; enactive approach; complex systems

---

## 1. Introduction

How can we design artificial agents such that their encounters with the world makes sense to them, that is, such that the meaningful aspects of those encounters are also experienced from their own intrinsic perspective as relevant? This is the problem of meaning, which has haunted artificial intelligence (AI) since the beginning of the field [1]. It has shown itself in different guises along the way [2], and it is essentially still with us today as more recent approaches to cognitive science continue the struggle to naturalize meaning [3]. It is important to be aware of this fundamental problem, especially nowadays when much of the general public, and surprisingly even many high-profile researchers, have been carried away by the current wave of high-profile advances in AI into believing that human-level AI is just around the corner, which would imply that this problem has finally been solved. We will argue that there are good reasons to remain thoroughly skeptical.

This is not the first time that our imagination has been dazzled by technical advances; in fact, history reveals that this occurs rather regularly [4]. Moreover, the perennial philosophical problem of how to make room for subjective meaning in a physical world, first articulated in its modern form by Descartes, remains fundamentally unsolved. The theory of mind that most researchers thought is the best bet to naturalize meaning, representationalism, has—after over half a century of concerted effort!—left it completely mysterious as to how mental content *as such* could make a difference in the

unfolding of natural processes [5]. Accordingly, there is a growing consensus that new approaches are required, especially in the form of non-representationalist embodied, dynamical, and enactive approaches, and one of us has been actively involved in developing such an enactive approach to AI [6]. But, as we will argue, these non-representationalist approaches to AI have ultimately inherited the same fundamental difficulty. Even the enactive approach, which has made a substantial effort to account for value and meaning in a non-representational manner [7], ultimately leaves it equally mysterious how the subjective, i.e. value, meaning, intention, purpose, etc., *as such*, on its own terms, could make a difference for the movements of an agent, if it is assumed that its internal and external activity is already completely governed by purely dynamical laws. Moreover, this assumption of state-determinism sits in tension with other commitments of the enactive approach that have received less attention in the literature, namely its insistence on the groundlessness and the interdependence of the subjective and the objective, which makes it possible to metaphorically conceive of action as "laying down a path in walking" [8].

We therefore propose that it is time to step back for a moment and ask ourselves whether the problem of this failure of naturalizing meaning perhaps not only derives from a faulty concept of *mind*, but rather from an inadequate concept of *nature*. Thus, following a brief review of the problem of meaning in AI and robotics, we discuss the conceptions of nature that are implicitly presupposed by representationalist and non-representationalist approaches to cognitive science. We then sketch an alternative concept of nature, one that places limits on its scope, and which we therefore believe has a better chance of accounting for the possibility that meaning makes a difference in a material world. We conclude by discussing the implications of this revised concept of nature for the design of artificial systems such that they make room for meaning to play a role.

## 2. The Problem of Meaning in AI and Robotics

As an illustration of the fact that the problem of meaning in AI is still very much with us today, consider the example of GoogLeNet, a convolutional neural network trained on more than a million images. It turns out that by adding an imperceptibly small perturbation to an image we can cause it to misclassify with high confidence such that, for example, an image that was classified correctly as a panda with 58.7% confidence becomes classified instead as a gibbon with 99.3% confidence [9]. The fact that deep neural networks are easily fooled in this and other ways [10], demonstrates that their otherwise impressive performance is not based on a meaningful perception of the content of their input. This continuing problem of meaning that is faced by current high-powered AI can have serious consequences, for instance in terms of issues surrounding the reliability of self-driving cars.

Over the years this overarching problem of meaning has been famously discussed in terms of a variety of more specific practical and theoretical problems, including the symbol grounding problem [11], the Chinese room argument [12], and generalizations of the frame problem [13,14]. A decade ago one of us co-authored an article [6] that diagnosed the root cause of this problem of meaning in AI as a lack of precarious self-individuation of artificial agents (i.e., as a lack of life, see also [15–17]). That article proposed as an alternative an enactive approach to AI that grounds artificial agency in both autopoiesis and adaptivity. At that time there were several well-known examples in the fields of cognitive robotics and artificial life of artificial systems that were either autopoietic [18] or adaptive [19], but not both. So the outstanding practical challenge of synthesizing an adaptive autopoietic system, plus compelling arguments derived from the philosophy of the organism that grounded intrinsic teleology and sense-making in self-regulated self-production [20–22], raised a tantalizing hope: that the problem of meaning in AI could finally be solved if we somehow managed to engineer the conditions for the emergence of adaptive autopoietic systems, at least at the level of habitual behavior or sensorimotor agency [23–25].

However, since then this hard problem of "second-order engineering" enactive AI [6], that is, of engineering a system that gives rise to an autopoietic individual that, in turn, during its agent-environment interaction gives rise to adaptive behavior, has essentially been solved, at least

virtually by using simulation models. Even the simplest models of reaction–diffusion systems can give rise to spatially individuated 'spots' capable of adaptive behaviors, including growing towards 'nutrient' sources and moving away from parasitic reactions [26,27]. A more complex example is a simulation model of a chemical system that supports a protocell that is capable of surviving certain kinds of perturbations by spontaneously adapting its internal chemical organization [28,29].

Following the enactive approach to value, according to which value is an emergent property of an autopoietic system's adaptive interactions [7], the protocell's tendencies toward its preservation are intrinsically good, whereas the ones leading to its disintegration are intrinsically bad. Thus, when the protocell rearranges itself internally and thereby avoids its own disintegration, we can say that it behaved according to an intrinsic norm which made this a good response. This notion of "behavior according to an intrinsic norm" is an essential element of the enactive definition of agency [30], and it has been given a more detailed dynamical treatment based on another simulation model of a chemotactic protocell [31,32].

But can we say that such a protocell's adaptive response to the perturbation is significant for the protocell itself? Does it have an intrinsic sense of having avoided something undesirable, bad, or imbued with negative valence? In other words, has the problem of meaning really been solved by combining autopoiesis with adaptivity? One worry, long highlighted by researchers working with real robots [33], is that simulation is not reality, and the possibility of transferring results from the former to the latter cannot be taken for granted. More specifically, the precariousness that grounds the concern inherent in living existence has no counterpart in a computer simulation whose entities are purely logical and hence essentially immortal [34].

But even in the case of real living beings, it only makes sense to claim that their meaningful perspective is grounded in their precarious existence if we accept, as Jonas [35] proposed, that their being is their own doing. An organism actively brings about its own physical existence, and it is this internal relationship between doing and being that makes the organism a being that is concerned with what it is doing, in particular with its continued self-preservation [36]. Of course, this is just an initial sketch and more has to be said about how the enactive approach could account for norms that are not related to the organism's need to avoid dying [37,38], for instance by developing detailed accounts of additional processes of self-generation that are hierarchically decoupled from metabolism [23]. Nevertheless, a fundamental worry would remain: to what extent are we justified in claiming that the organism's being is something actively done, rather than merely passively undergone, if all of its unfolding processes are completely prespecified by a deterministic universe? Such a universe is not compatible with Jonas' characterization of life as "needful freedom" [20]. A precarious existence may be necessary for a meaningful perspective, but so is the freedom to make a genuine difference with respect to the needs of this existence.

We will not enter more deeply into the enactive approach to grounding meaning here, because our primary concern lies with how meaning, once present, could make a difference for behavior in the physical world, no matter how meaning arises in the first place. Thus, for us an even bigger worry is that even if we were to attribute some minimal intrinsic sense of normativity to a model protocell, or a living being in a deterministic universe, then this normativity would for all practical purposes be essentially irrelevant for its adaptive behavior, which may be emergent, but which ultimately is still completely determined by the system's dynamical laws. It is therefore in principle possible to give a complete dynamical account of all of the agent's activity, and we can identify trajectories that lead to disintegration and others that lead to adaptation. The key worry is that, given that a complete dynamical description of the system's activity is indeed possible, there is no longer any conceptual room for the agent's norm *as such* to make any difference to the behavior. The agent adapts to some perturbations not because it is somehow concerned about its continued existence, but just because it is absolutely determined to do so by the overall chemical system's dynamics.

To put it differently, while at first sight the notion of "behavior according to an intrinsic norm" suggests that an agent behaves the way it does because that is what is good for it, it is actually more

correct to say that is behaves that way simply due to certain dynamical constraints on its internal and interactional dynamics. Whether we label a region of this dynamical state space as being in line (or in tension) with its norms simply makes no difference for the agent, nor to the unfolding of its trajectories. Once the agent has started on a particular trajectory, it can never leave that trajectory—even if it is trajectory that is supposedly intrinsically bad for it. To repeat, in this framework the norm as such makes absolutely no difference to activity: the behavior is something that is just undergone by the system, rather than actively chosen to be in accordance with an intrinsic norm.

This dynamical interpretation of emergent normativity in an adaptive autopoietic system, which should not be controversial for those enactivists who are still committed to state-determinism, places the protocell in the same class as, let us say, the moon whose trajectory in outer space is constrained by the gravitational pull of the earth. One the one hand, this dynamical equality between living bodies and mere objects is good news, since this would imply that the naturalization of normativity has been achieved. However, this scientific victory of a strict naturalism comes at the considerable cost of making normativity, and subjectivity more generally, fundamentally ineffective in nature. This consequence is especially highlighted in Maturana's original formulations of autopoietic theory [39], which still leads some to accept the eliminativist claim that living beings are "essentially machines composed of chains of deterministic processes. Everything that the system does is determined by its structure at a given moment, not goals or desires about states in the future and not selecting between features of the environment that harm or benefit the organism" [40] (p. 666).

We believe that this impotence of subjectivity is too steep a price to pay for the naturalization of meaning, and we are therefore motivated to explore alternatives. Ideally, we would like a theory of the living that leaves room for meaning and intentional action to make a difference on their own terms in the natural world. Otherwise the theory is in direct tension with lived experience: when I act normatively I have the experience that I act this way because I chose to satisfy the norm, and not because I am simply forced to undergo the behavior [41]. For instance, we wrote this essay because we believe its arguments to be correct and valuable. But if our writing behavior is actually completely determined by physical causes, it would mean that this kind of meaningful experience of doing something for reasons is nothing but an illusion, a misleading epiphenomenon. We would feel free from physical determination only because of our ignorance about the ultimately unfree causes of our behavior. This is a pessimistic conclusion that is acceptable to many (e.g. [42]), but it should be unacceptable for an enactive approach that aims to genuinely heal the division between cognitive science and human experience [43,44].

This critical assessment of the state-of-the-art of AI and artificial life suggests that the proposal to solve the problem of meaning by creating artificial agents that are both autopoietic and adaptive has to be revised. We believe that autopoiesis and adaptivity continue to be important concepts for the naturalization of meaning, but they do not provide the full story: theoretical room has to be made such that the normativity of an agent can also make a difference for its unfolding activity in terms of that normativity as such. As we will argue in the next section, this shortcoming can motivate us to go beyond the current focus on the individual agent and to consider an enactive theory of meaning, and of subjectivity more generally, that involves revising the concept of nature as a whole.

## 3. Varieties of Naturalization

There are two prominent kinds of explanatory strategies for the naturalization of mind [45]. The dominant strategy is representationalism, which involves accounting for meaning's place in nature in terms of mental content, which is typically done by positing some extra mediating factor in the brain of the agent, namely representational vehicles that intrinsically carry representational content. An alternative strategy is to account for meaning's place in nature directly in terms of neural and/or behavioral dynamics, such as order parameters and phase transitions that guide an agent's behavior in relation to environmental features. There are many similarities and also important differences between these two naturalization strategies, but arguably the most fundamental difference is whether or not

they add some extra element to nature in order to account for meaning. We will therefore label them "nature++" and "nature==", respectively.

### 3.1. Nature++

Traditionally, AI and cognitive robotics has been inspired by the representationalist strategy of naturalization in its attempt to synthesize artificial minds [46], which we call the *nature++* strategy because it seeks to add mental content into nature. Representations in classical AI and some forms of embodied AI can take many forms, such as knowledge bases, cognitive maps, and value systems. As we already mentioned in the introduction, this naturalization strategy has run into a wide variety of practical and theoretical problems that can all be seen as different expressions of the problem of meaning. Simply put, a computer only operates in terms of the logic of its syntax, and it makes no difference to that operation what the states of the variables are supposed to represent, or whether they mean anything in the real world at all [47]. This independence of semantic content from syntax is not all bad, because it entails that the same formal algorithm can be used to solve different practical problems. From the observer's perspective the same algorithm or modeled dynamical system could be interpreted as simulating a chemical reaction–diffusion system, or predatory and prey dynamics, or as nothing but abstract coupled equations. But, crucially, whether a logical expression such as $x = 0.2$ refers to the concentration of reactants or of rabbits makes no difference to the code's execution.

It might be thought that this problem of representational content does not apply to connectionist AI, and that therefore the current advances in deep artificial neural networks are exempt from this difficulty. However, despite its preference for sub-symbolic architectures, connectionism is part of the computational theory of mind broadly conceived [48], and it therefore inherits a version of the problem of meaning. When explaining the operation of a neural network, dynamical concepts such as basins of attraction and bifurcations will be entirely sufficient. To return to an earlier example, it simply makes no difference whether an attractor in the network's dynamics represents images of pandas or of gibbons. This irrelevance of representational content for computation also holds for the explanation of brain activity. While it is certainly possible for scientists to ascribe content to patterns of neural dynamics, such as working with the activations of a certain brain region as if they were a representation of the agent's location or orientation in space, no one has been able to show how such representational content *as such* could make a difference to the neural activity [49]. Neuroscientists studying individual neurons have not yet been forced to appeal to anything other than completely natural processes, that is, processes known from the rest of nature such as changes in chemical substances and electrical potential, to explain how and when a neuron will fire.

Given this apparent impotence of the content of representations for the operation of a digital computer and of the brain, it is not surprising that many researchers avoid appeals to representational content altogether. Instead they prefer a non-representationalist strategy.

### 3.2. Nature==

Non-representationalist approaches to AI and cognitive robotics famously started to take off in 1990s. Examples include behavior-based robotics [50], the dynamical approach to cognition [51], and certain strands of evolutionary robotics [52]. While the rejection of internal representations can lead to a kind of dynamical eliminativism, a more popular approach has been to directly identify aspects of mind with patterns of the brain, or with whole brain-body-world dynamics. Philosophically, we can conceive of this approach as a suitably updated version of identity theory [53], which is why we will refer to it as the *nature==* strategy. Its key claim is that mind should be identified with emergent properties of neural and/or behavioral dynamics that in turn exert top-down constraints on those same dynamics. Different versions of this claim have been elaborated by a whole range of non-representationalist approaches to cognitive science [54–57]. For example, it has been developed into a scientific research program by Kelso and colleagues, who account for mind in terms of emergent order parameters. This allows them to identify meaningful actions with constraints that shape the

agent's internal dynamics: "In coordination dynamics, intention acts in the same space as the intrinsic dynamics, attracting the system toward an intended pattern. Intentions constrain and are constrained by the intrinsic dynamics. They may both stabilize and destabilize patterns of behavior" [58] (p. 222). On this view, in nature there are just different kinds of dynamics, and nothing else.

We take seriously the starting point of the nature== strategy that there is as yet no evidence for representational content playing any role inside nature. In this sense, the physical realization of meaningful actions lacks something: Maturana is right that from a third-person perspective we do not find the intentions and goals we experience from a first-person perspective. The physical basis of meaning does not stand apart from the rest of nature; it does not carry any mark of the subjective. In this sense nature can be said to be essentially incomplete, as Deacon [59] has highlighted. However, Deacon's appeal to incompleteness does not entail a rejection of physicalism because the non-representationalist strategy of nature== remains available: on this view, meaning just *is* a certain kind of pattern of constraints that is nowhere to be found concretely and yet this pattern still effectively governs activity.

However, while this strategy removes the problem of how representational content could play a role by denying the existence of content in nature, it does not manage to solve the fundamental problem of how meaning as such could make a difference on its own terms, over and above the lawful dynamics, and so we are at best left with another version of epiphenomenalism. Even worse, while the nature++ strategy is unable to shake the worry that representational content is superfluous for our best scientific accounts of mind, the nature== strategy further raises the stakes by fostering the possibility that upon closer inspection mind itself may in the end collapse into nature and become conceptually eliminated.

### *3.3. Nature–*

Given that the nature++ and nature== strategies give rise to the problem of mental impotence, i.e. the incapacity of the subjective to make a difference over and above the objective, and assuming that it is desirable to aim for a naturalization of the mind that leaves room for a role of the subjective in its own right, a different strategy seems to be called for. Both strategies in their own way suffer from an overdetermination of meaningful action, in which behavior is conceived of as saturated by objective determination to such an extent that no conceptual room is left for the subjective as such. How can we make the necessary conceptual room? More precisely, how can we accept on the one hand that there is no positive evidence for mind in nature, and yet at the same time accept that meaning as such makes a genuine difference in the unfolding of natural processes?

Our proposal is to go even further than the nature== strategy: instead of only rejecting the quest for evidence of the *presence* of the *subjective* in nature, we should rather look for evidence of the *absence* of the *objective* in nature. In other words, we propose that when subjectivity is viewed from an external, third-person perspective, it is forced to appear as objective and therefore cannot appear as subjective directly, but it can nevertheless manifest itself indirectly through a relative lack of the normally expected physical determinations compared to those of an ordinary object [60].

The upshot is that while we can try to approximate living beings as state-determined systems, and even do so with some success, ultimately, their irreducible subjective nature prevents this partial approximation from being turned into a complete specification. Moreover, we argue that this is not a practical limitation due to measurement problems or a lack of knowledge. Rather, if our proposal is correct, this is a necessary limitation when approaching the subjective via an objective perspective, and as such the limitation can only ever be fully overcome at the price of subjectivity itself: A living being can never be fully known objectively as long as it is living, but only when it is dead and hence no longer behaves in accordance with subjective norms. Nevertheless, this in principle limitation of complete objective determination should itself be objectively detectable in practice. For example, if we could adapt the classical two-slit experiment of quantum physics to living beings, such that a living being has to pass between one of two exactly identical slits in a wall, we predict that, just like in the

original particle version, it would be in principle impossible to predict the outcome on a trial-by-trial basis, no matter how simple the organism and no matter how much we know about the situation. This kind of experiment would then allow us to try to better understand the processes that give rise to this uncertainty at the behavioral level, i.e. those which make room for the subjective.

Particularly fitting candidates for studying this intertwining of the subjective and the objective would be natural phenomena that are physically incomplete systems and whose activity is causally underdetermined by preceding physical events inside and outside of the system. A good starting point might actually be to reconsider the autopoietic system: metabolic self-production entails a self-reference at the core of its being such that it is never fully self-coinciding at any one moment in time, and this intrinsic circularity makes it formally equivalent to an incomplete system [61]. However, we would have to overcome the determinism assumed by current models of autopoiesis.

Note that nondeterminism is not the same as randomness, as randomness can be generated in a deterministic manner, e.g. by using a look-up table containing random numbers [62]. Moreover, non-deterministic behavior can still have some structure over time, albeit a kind of structure that is only temporarily constrained but never fully determined by preceding physical state. To make room for such nondeterministic behavior, the overarching system that contains the autopoietic system cannot be a completely deterministic system, either, which seems to rule out the possibility of using standard computational simulation. Making this room also seems to rule out the idea that nature is causally closed and that it will bottom out at some smallest physical scale, and instead suggests that nature is nonergodic [63] and groundless [64].

We therefore require a concept of nature that is incomplete in a deeper way than simply lacking evidence of mental content or a mark of the subjective. We must give up the idea of the causal closure of the universe and accept incompleteness and indeterminacy as essential properties of nature in order to make room for the subjective in the objective. We will refer to this strategy as nature–.

To many it will seem strange that nature should be conceived of as essentially nondeterministic and incomplete, but it should be remembered that there is nothing about the concept of nature that would in principle disallow such properties. To the contrary, we find that nature is nondeterministic at the quantum level [62]. Similarly, that the phenomenon of entanglement cannot be explained in traditional terms of cause and effect, which is why Einstein referred to it as spooky action at a distance and claimed that quantum mechanics must be an incomplete theory, is now simply accepted as a brute fact about nature. In other words, nature is already sufficiently strange at its core. The nature– strategy simply amounts to also making conceptual room for nondeterminism and incompleteness at the scale of everyday physical objects, specifically that of living bodies.

To illustrate this possibility, it is helpful to remember that ethology and psychology have in practice already developed effective methods of working with the uncertainty of the behavior of the living, namely in terms of probability functions. The key theoretical difference, from the perspective of the nature– strategy we are proposing, is that it is misguided to explain away all of this uncertainty by attributing it to practical limitations of current scientific practices. To the contrary, if we are correct in claiming that this uncertainty is in fact a result of the indeterminacy through which the subjective can express itself in objective terms, no future or even idealized scientific practice would be able to eradicate it completely. Accordingly, the nature– strategy may provide a promising starting point from which to develop an alternative theory of agency and behavior that can make better sense of psychology's widely publicized replication crisis.

To further motivate this strategy, it is worth considering that demonstrations of quantum effects are slowly being scaled up to larger objects [65], and it is not clear yet at which scale they will stop. For the case of quantum phenomena, it is already accepted that the uncertainty is inherent in the phenomena, rather than resulting from methodological inaccuracies. But while the nature– strategy does not rule out quantum effects directly playing a role at the scale of the living, as suggested for example by Kauffman [63] (p. 150), it is also open to the even more radical possibility that life has its own way of being nondeterministic and incomplete.

From a complex systems perspective on life and mind, this idea is not that far-fetched. It is already widely accepted that animal life is characterized by structured but empirically unpredictable behavior, which has been approximated in terms of self-organized chaos [66], metastability [67], and self-organized criticality [68]. Accordingly, it is but a small conceptual step to add nondeterminism and incompleteness to this existing mix[1]. Moreover, in practice it will be difficult to experimentally distinguish between nondeterministic behavior and deterministic but unpredictable behavior, such as behavior arising from deterministic chaos. For instance, a study of the spontaneous movement trajectories of a fly revealed that its behavior could be partially approximated in terms of deterministic embodied chaotic itinerancy, but that at times it also seemed to exhibit intrinsic randomness [71].

If life really is nondeterministic then it might be more productive to stop trying to capture its behavior with deterministic models, even of the chaotic kind, and simply capture the irreducible uncertainty of behavior in probabilistic terms, as has been done for the quantum level in a highly successful manner. Intriguingly, there is increasing evidence that many aspects of human decision making are better captured by quantum models compared to standard probabilistic models, giving rise to a field known as quantum cognition [72]. The nature– strategy suggests that this approach is on the right track and could be generalized to quantum life. However, this does not mean that we advocate the abandonment of existing work on deterministic unpredictability. We speculate that the fact that the brain is highly sensitive to initial conditions and is poised at criticality, is a way of amplifying the uncertainty that otherwise lies dormant at the core of nature. For instance, Jonas [73] speculated that living bodies are precariously poised in a state of unstable equilibrium, like an upside-down cone centered on its tip, such that any infinitely small, and hence practically unobservable, perturbation would lead to a bifurcation in the system's dynamics, thus bringing about a new macroscopic state configuration (e.g. the cone will fall sideways, although it is impossible to predict in which direction). In this way the indeterminacy inherent in nature would be amplified by the living body's degrees of freedom, an amplification which reaches astronomic scales in the human case when we consider that each of our brain's 86 billion neurons constitutes a degree of freedom in the nervous system's dynamics [74].

Future work should investigate in more detail the possible underlying bases for this amplified nondeterminism and incompleteness in the behavior of the living. Is it just a brute fact of nature that living bodies are characterized this way? And will no further understanding be forthcoming, similar to the current mainstream physics stance regarding the strange nature of quantum mechanics? But this simplistic equation with quantum physics would overlook the fact that at the macroscopic level of living bodies a physically nondeterministic behavior can still be a meaningfully determined action. For example, that I reach for my cup can be physically underdetermined even if we knew the entire history of the physical universe, and yet I could also tell you that I reached for it because I wanted to drink my coffee. Accordingly, as Jonas noted, quantum indeterminacy contrasts with behavioral indeterminacy because in the latter there is another source of evidence: In both cases we are faced with an event that is physically underdetermined when observed from a third-person perspective, but in the case of my own behavior I can also experience it from the first-person perspective. From that perspective I know that the behavior is an intentional action done by myself for reasons that go beyond the physical conditions that precede it, that is, in accordance with norms. We are aware that this first-person recognition of the role of normativity in acting does not amount to an explanation of how a subjective intention can make a difference for objective movement in practice, but at least this recognition accepts that an intention could make a difference in principle.

---

[1]　More specifically, we argue that this is a small conceptual step for a scientific perspective that is already used to dealing with complex phenomena that are inherently unpredictable. Nevertheless, we acknowledge that this step has profound implications for our understanding of reality that deserve to be more fully developed in future work, for instance by taking inspiration from related work in the philosophy of physics [69,70].

On this view, the fact that the human brain gives our physical indeterminacy the largest degrees of freedom known in the animal kingdom, at least when the number of neurons is controlled for body size [74], is the physical complement of our unprecedented subjective capacity for volition, that is, the basis for human freedom and responsibility. We still do not understand just how the subjective takes advantage of the openings in the objective to physically exert its will in this manner. But a good place to start is to simply acknowledge the very possibility of this increasingly complex intertwining relationship, in which the subjective and the objective make room for each other. As Thompson puts it: "the cognitive complexity of consciousness increases as a function of the increasing complexity of living beings. Consciousness depends on physical or biological processes, but it also influences the physical or biological processes on which it depends" [75] (p. 103). We are in agreement with Thompson that a proper understanding of how subjectivity is capable of making this difference to how physical processes unfold will require further revisions of the concept of nature, specifically "a nondualistic framework in which physical being and experiential being imply each other or derive from something that is neutral between them" (ibid.: 105). To make sense of this interdependence between mind and world will not be easy, but we believe that contemporary phenomenology and the enactive approach are poised to come together to develop the appropriate philosophical framework [44,76]. The enactive approach has shown the way toward a nonreductive naturalization of phenomenology, and it is now phenomenology's turn to help us return to the things themselves so that we can develop a complementary nonreductive phenomenologization of nature [77][2].

## 4. Discussion

If this nature– strategy is on the right track, then it would mean that digital computers and classical dynamical systems more generally are inherently unsuitable frameworks for embodying meaning, given that they are complete and deterministic systems. The upshot is that if we want to design artificial systems that solve the problem of meaning, then we have to build them such that their objective determinations (as expressed in terms of systemic completeness, causal closure, state determinism, etc.) can partially withdraw so as to make room for subjective influences to be able to make a difference in their own right. Only in this way could an artificial system become a suitable medium for the embodiment of an agent for whom things show up as meaningful.

In order to design such artificial systems, we do not need to directly create genuine freedom in such systems, which is too difficult and overly adventurous. Instead, our proposal is much more modest. What we have to do is only to take seriously that indeterminacy is already dormant in nature itself, and to find a way to amplify the indeterminacy that already exists in a meaningful manner. In order to do that systematically it is necessary to learn from life and to better understand how the brain could intensify such indeterminacy toward meaningful behavior at the macroscopic level. The key question would therefore be how to liberate this fundamental, primitive indeterminacy in microscopic nature from the general macroscopic tendency toward stable orders and patterns, i.e., how to build a mechanism to cancel or counteract such macroscopic trend of nature at particular, localized, limited, individual focal points. Such a locally intensified negation of the macroscopic tendency toward stability could create the necessary opening for the subjectivity of an agent.

We can already discern some first steps in this direction in the current literature on AI and robotics. For representationalist AI it would be important to extend the classical Turing machine in a direction that would make indeterminacy an intrinsic aspect of its operation, perhaps in terms of entropy arising from distributed computations [80], or perhaps by taking advantage of decoherence and other noise

---

2    We focus here on the contributions of phenomenological philosophy because we are most familiar with that tradition. Yet we certainly recognize that there are other traditions that have much to offer for the development of a suitably revised concept of nature, including the speculative naturalist philosophy going back to Peirce and Whitehead, as well as contemporary movements within analytic philosophy that argue for more "liberal" [78] and "relaxed" forms of naturalism [79]. Future work could compare and contrast these diverse proposals.

in quantum computing. Initial steps toward incorporating quantum effects into artificial life have also been reported [81]. This possibility of taking advantage of the indeterminacy of hardware also puts in new light the insistence of some proponents of embodied cognition on using physical robots as a testbed for theories of cognition [82]. Even without going to the quantum level, the physical limitations of robots can be harnessed to produce more life-like behavior: "physical constraints in time and space do not allow the system to be uniquely optimized and thus give rise to incompleteness and inconsistency. Actually, in our robot experiments, such inconsistencies arise in every aspect of cognitive processes including action generation, recognition of perceptual outcomes, and the learning of resultant new experience." [83] (p. 252). Similarly, in artificial life there are indications that we need to complement clean, closed, "minimal" simulation models of cognition [84] with more messy, open, "massive" or "maximal" physical life-like systems built in hardware [85] or even wetware [86,87].

While these synthetic approaches could lead to the generation of artificial systems that leave room for subjective meaning to make a difference, just the existence of this opening per se does not entail that meaning will make a difference in practice. In other words, we suspect that incompleteness and indeterminacy are necessary but not sufficient conditions for meaning to be effective. At this point it is not clear what other conditions are required for obtaining sufficiency. One possibility is that it requires some kind of involvement of life itself: since the origin of life nearly four billion years ago, life only comes from life, which would mean that in nature at least the subjective is always already directly participating in the generation of new manifestations of the subjective. This suggests that a potentially more fruitful synthetic approach to address the problem of meaning could be to incorporate existing manifestations of the subjective into artificial systems. We can see precursors of this idea in the cybernetic period, especially in Beer's and Pask's attempts to harness the complex dynamics of animals or whole aquatic ecosystems in adaptive controllers [88] (pp. 231–234). They did not succeed, but today's cognitive robotics has started to work again on the incorporation of much simpler living organisms into its designs, namely bacteria and neural cells [89,90]. Neurons in particular could be more malleable for this purpose [91].

At the other end of the scale of organismic complexity we find humans, which brings us to the whole field of human–computer interaction. Indeed, one failsafe way of practically solving the problem of meaning in AI and robotics is to make sure that there is always a human somewhere in the behavioral loop. At least in the near future, if the nature– strategy is on the right track, researchers interested in using a synthetic approach to generate technological advances based on meaning would be better served shifting their focus from duplicating human understanding in artificial systems, to directly empowering humans by extending their existing subjective capacities by designing better interfaces [92]. This would calm misplaced worries about human-like AI taking over the world, and instead refocus attention and resources on designing interfaces for steering the artificial large-scale complex adaptive systems that already dominate our lives [93].

**Author Contributions:** Conceptualization, T.F. and S.T.; writing—original draft preparation, T.F.; writing—review and editing, T.F. and S.T.

**Funding:** This research received no external funding.

**Conflicts of Interest:** The authors declare no conflict of interest.

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
