# Peer review of "The Problem of Meaning in AI and Robotics: Still with Us after All These Years"

_philosophies, doi:10.3390/philosophies4020014_

Round 1
Reviewer 1 Report
This article makes a clear and compelling case that a number of persistent problems in AI and cognitive science -- gathered together as "the problem of meaning" -- cannot be addressed without reconstructing our view of nature. It briefly argues that existing strategies -- adding representational content, or treating mind as just another dynamical system -- do not work. The authors then argue for a nondeterministic concept of nature that can "make room" for subjective influences. I agree that this is a necessary step, and I also agree with the authors' view that it is insufficient: a nondeterministic nature does not give us a naturalistic concept of how reasons and norms act as causes. But these are very difficult questions, and I think that the authors make an important contribution simply by raising them.
I have two suggestions for minor alterations, which I leave to the discretion of the authors.
First, I urge the authors to reconsider their statement that "it is but a small conceptual step to add nondeterminism and incompleteness to the mix" (line 284). I suggest that, in fact, the full implications of this step are revolutionary. For one, it implies that time is real. For a similar perspective, see the recent works of the physicist Lee Smolin, especially his Time Reborn (2013) and the manifesto that he wrote with the philosopher Roberto Unger, The Singular Universe and the Reality of Time (2015). The addition of incompleteness is not a minor alteration to an otherwise comprehensive and adequate picture of nature.
Second -- for future reference, but perhaps not for this paper -- I suggest that the authors look at serious work that has been already been done by philosophers on what they are calling the "phenomenologization of nature." There is a tradition of speculative naturalist philosophy running back to Peirce and Whitehead that has been engaged in this task for more than a century. Also, in recent years, some analytic philosophers have begun to venture similar kinds of arguments: see especially Gregg Rosenberg, A Place for Consciousness: Probing the Deep Structure of the Natural World (2004). I think that enactivists a lot to add to these conversations about the reconstruction of nature, but first they have to acknowledge that that these conversations already exist.
Author Response
This article makes a clear and compelling case that a number of persistent problems in AI and cognitive science -- gathered together as "the problem of meaning" -- cannot be addressed without reconstructing our view of nature. It briefly argues that existing strategies -- adding representational content, or treating mind as just another dynamical system -- do not work. The authors then argue for a nondeterministic concept of nature that can "make room" for subjective influences. I agree that this is a necessary step, and I also agree with the authors' view that it is insufficient: a nondeterministic nature does not give us a naturalistic concept of how reasons and norms act as causes. But these are very difficult questions, and I think that the authors make an important contribution simply by raising them.
AUTHORS:We thank the reviewer for these encouraging remarks. Our primary purpose in this contribution is indeed to help clarify these profoundly difficult problems and to raise awareness about possible alternative paths forward.
I have two suggestions for minor alterations, which I leave to the discretion of the authors.
First, I urge the authors to reconsider their statement that "it is but a small conceptual step to add nondeterminism and incompleteness to the mix" (line 284). I suggest that, in fact, the full implications of this step are revolutionary. For one, it implies that time is real. For a similar perspective, see the recent works of the physicist Lee Smolin, especially his Time Reborn (2013) and the manifesto that he wrote with the philosopher Roberto Unger, The Singular Universe and the Reality of Time (2015). The addition of incompleteness is not a minor alteration to an otherwise comprehensive and adequate picture of nature.
AUTHORS:We added a footnote (footnote 1) to clarify our intention and to add a citation to Time Reborn: “More specifically, we argue that this is a small conceptual step for a scientific perspective that is already used to dealing with complex phenomena that are inherently unpredictable. Nevertheless, we acknowledge that this step has profound implications for our understanding of reality that deserve to be more fully developed in future work, for instance by taking inspiration from related work in the philosophy of physics [69,70].” (p. 8)
Second -- for future reference, but perhaps not for this paper -- I suggest that the authors look at serious work that has been already been done by philosophers on what they are calling the "phenomenologization of nature." There is a tradition of speculative naturalist philosophy running back to Peirce and Whitehead that has been engaged in this task for more than a century. Also, in recent years, some analytic philosophers have begun to venture similar kinds of arguments: see especially Gregg Rosenberg, A Place for Consciousness: Probing the Deep Structure of the Natural World (2004). I think that enactivists a lot to add to these conversations about the reconstruction of nature, but first they have to acknowledge that that these conversations already exist.
AUTHORS:We did not want to give the impression that phenomenology is the only way forward in terms of future developments of our proposal. We therefore added another footnote (footnote 2) to clarify this point: “We focus here on the contributions of phenomenological philosophy because we are most familiar with that tradition. Yet we certainly recognize that there are other traditions that have much to offer for the development of a suitably revised concept of nature, including the speculative naturalist philosophy going back to Peirce and Whitehead, as well as contemporary movements within analytic philosophy that argue for more “liberal” [78] and “relaxed” forms of naturalism [79]. Future work could compare and contrast these diverse proposals.” (p. 9)
Reviewer 2 Report
1- How I read the paper:
This is a fine work. The authors were able to convey their view well and they also articulate their thesis well. Although their main argument is not very strong, they propose an interesting strategy for re-conceiving the notion of nature and briefly indicate how this view would be advantageous to advances in AI.
Below, I’ll explain my understanding of the paper and suggest improvements.
First, the authors provide an overview of the problem of meaning. Then, they try to show that recent conceptions arguing in favor of adaptive autopoietic systems as sufficient for meaning are inconsistent. Subsequently, they use this framework - of the insufficiency of adaptive autopoietic systems for meaning - to propose the necessity of a reconsideration of the notion of nature as the key element for overcoming the problem of meaning in AI.
This is justified by them by the lack of space for indeterminacy in adaptive autopoietic systems, which would be a requirement for meaning.
In order to present their proposal, they consider two other perspectives on the relation between nature and cognitive systems, the addictive one (nature++) and the identity one (nature==). ‘Nature++’ stands for representationalist theories which pose an extra element in between the cognitive system and the environment.‘Nature==‘ stands for enactive theories that identify processes of the mind with brain-body-world dynamics.
The authors, then, propose a subtractive perspective: Nature--. This conception is supposed to create room for the concept of mind in nature. This perspective, in my understanding, is a clever attempt to dissolve the duality between objectivity and subjectivity by explaining that subjectivity is forced to be objective when it is seen from a third person perspective. From there, the proposal is to look for the absence of the objective in nature.
They argue that the best way to avoid determinism, and therefore give conceptual room for meaning, is to opt for quantum theories of cognition, which can capture the uncertainty of behavior in probabilistic terms.
2- The problems I found on the current version of the text:
(Changes required)
2.1- Conflating different problems of meaning into one.
The symbol grounding problem is the problem of “How can the semantic interpretation of a formal symbol system be made intrinsic to the system” [1]. While the Chinese room thought experiment addresses precisely the question whether machines can understand semantic content, the frame problem is a more specific problem of how to limit the context of propositions, and in its formal aspect, it is considered fairly solved [2].
Suggestion: On lines 69 and 70, use a formulation more similar to lines 171 and 172: “a variety of practical and theoretical problems”
2.2- Imprecise terminology about enactivism.
Not all enactive theories are non-representationalist. Only the radical views abandon the notion of representation.
Suggestion: revise use of terminology when it comes to 4E cognition. Make it clear what are the differences with the radical views.
2.3- Conflating the notion of agency with attribution of meaning.
The notion of agency developed by Barandiaran et, al. [3], does not claim that agency entails semantic understanding.
There is an important notion when it comes to Barandiaran at.al.’s definition of agency, that is neglected by the authors, namely, asymmetry. This notion explains the difference between the proto-cell example and a system that can be considered an agent.
Asymmetry, in my understanding, could also give room to indeterminacy, which would be a feature of the agent’s actions. But this is debatable.
Suggestion: consider the conceptual role of the notion of asymmetry.
2.4- How would you define subjectivity in Nature--?
The idea that the absence of objectivity in nature gives room for meaning is not straight-forward. If I understood it correctly, keeping the idea that meaning is given in subjectivity wouldn’t work after the dissolution of the subjective vs. objective duality.
Suggestion: address the question of how the absence of objectivity will leave room for meaning more directly.
2.5- Specific Lines:
47 - What is meaning as such? Perhaps make reference to semantic content, maybe in a footnote?
245 - What is a relative lack? Perhaps explain relative to what?
316-320 - This explanation seems too short.
3- Corrections, optional suggestions, and discussion points:
(Optional)
3.1 Lines:
14 - replace ‘show up’ for ‘appear’
46 - replace ‘the enactive approach’ for ‘radical enactive approaches’
183 - correct: ‘a sub-symbolic architectures’
191 - correct: representation ‘of’ the agent’s…
217- In this sense (comma)
233 - replace ‘mental impotence’ for a more precise expression
241-246 - Give an example
273-275 - Give an example
309-311 - Sentence is too long and confuse
313 - delete ‘for’ (I grasp my cup)
318 - replace ‘can I’ for ‘I can’
318 - delete ‘my’ (from the first-person perspective)
284- replace ‘And so it is’
3.2 - It seems to me that if we consider nature as essentially non-deterministic and incomplete, the interaction between organism and environment ceases to be determined not due to the agent’s ‘freedom’ - which is a requirement for meaning - but due to the very nature of nature. How would this affect your notion of meaning?
3.3 - I had the impression that you first opt for dissolving the duality between subjectivity and objectivity (lines 243-245), and later on you use the notion of subjectivity to justify meaning from the first person perspective (317-320). If the aim is to replace the duality with something neutral between them (334), how would meaning be subjective?
Another question is: How could I see meaning in the behavior of others?
3.4 - I can imagine an artificial agent who makes decisions, based on quantum models, which give room for indeterminacy, and still doesn’t grasp the meaning of its actions. I see that at the end of the paper you mention that indeterminacy would not be sufficient, but I’m not quite convinced that it would be necessary either. Perhaps there is a fine distinction on the notion of indeterminacy that we are not contemplating.
When would you say that a ‘natural’ agent (someone) understood a rule, or grasped the meaning?
I can also imagine an agent that is not free and yet understands its actions. (I cannot avoid acting the way I act, nevertheless I know why I act this way.)
Is being free a necessary condition for understanding?
3.5- The main argument, namely, that a re-conception of the notion of nature should take place, is based on the idea that adaptive autopoietic systems are insufficient for meaning. If we reduce the argument into simple steps it would be like this:
We need to understand meaning in order to advance in AI
The conception of adaptive autopoietic system is not sufficient to account for meaning
Therefore we should reconsider our notion of nature
Of course, this reduction is neglecting several steps, but based on this impression that I had from the text, I would suggest making it clear that your suggestion of a re-conception of the notion of nature doesn’t follow from the fact that the concept of adaptive autopoietic systems is insufficient for meaning.
4- Overall Recommendation
Accept - revisions required.
I suggest the authors consider the points mentioned in item 2. Suggestions on item 3 are optional to improve readability. Discussion points are also optional.
5- References
[1] Harnad, S. (1990). The symbol grounding problem. Physica D: Nonlinear Phenomena, 42(1), 335–346. https://doi.org/10.1016/0167-2789(90)90087-6
[2] Shanahan, M. (2016). The Frame Problem. In E. N. Zalta (Ed.), The Stanford Encyclopedia of Philosophy (Spring 2016). Metaphysics Research Lab, Stanford University. Retrieved from https://plato.stanford.edu/archives/spr2016/entries/frame-problem/
[3] Barandiaran, X. E., Di Paolo, E., & Rohde, M. (2009). Defining Agency: Individuality, Normativity, Asymmetry, and Spatio-temporality in Action. Adaptive Behavior, 17(5), 367–386. https://doi.org/10.1177/1059712309343819
Author Response
1- How I read the paper:
This is a fine work. The authors were able to convey their view well and they also articulate their thesis well. Although their main argument is not very strong, they propose an interesting strategy for re-conceiving the notion of nature and briefly indicate how this view would be advantageous to advances in AI.
Below, I’ll explain my understanding of the paper and suggest improvements.
First, the authors provide an overview of the problem of meaning. Then, they try to show that recent conceptions arguing in favor of adaptive autopoietic systems as sufficient for meaning are inconsistent. Subsequently, they use this framework - of the insufficiency of adaptive autopoietic systems for meaning - to propose the necessity of a reconsideration of the notion of nature as the key element for overcoming the problem of meaning in AI.
This is justified by them by the lack of space for indeterminacy in adaptive autopoietic systems, which would be a requirement for meaning.
In order to present their proposal, they consider two other perspectives on the relation between nature and cognitive systems, the addictive one (nature++) and the identity one (nature==). ‘Nature++’ stands for representationalist theories which pose an extra element in between the cognitive system and the environment. ‘Nature==‘ stands for enactive theories that identify processes of the mind with brain-body-world dynamics.
The authors, then, propose a subtractive perspective: Nature--. This conception is supposed to create room for the concept of mind in nature. This perspective, in my understanding, is a clever attempt to dissolve the duality between objectivity and subjectivity by explaining that subjectivity is forced to be objective when it is seen from a third person perspective. From there, the proposal is to look for the absence of the objective in nature.
They argue that the best way to avoid determinism, and therefore give conceptual room for meaning, is to opt for quantum theories of cognition, which can capture the uncertainty of behavior in probabilistic terms.
AUTHORS:We thank the reviewer for this careful reading of our contribution. We would only like to clarify that our aim is not “to dissolve the duality between objectivity and subjectivity”, but to work toward a concept of reality in which both can coexist in an interdependent manner. Moreover, our project in this paper is more methodological than metaphysical: we highlight that when subjectivity is approached with the scientific perspective of objectivity by definition it cannot appear as subjectivity, but it could still appear as the negation or bracketing of the kind of objectivity that would normally apply to macroscopic objects. The formalisms of quantum physics could therefore be suitable to capture the inherent indeterminacy of life and mind to the extent that these formalisms were designed to describe other phenomena that also defy the characteristics of macroscopic objects.
2- The problems I found on the current version of the text:
(Changes required)
2.1- Conflating different problems of meaning into one.
The symbol grounding problem is the problem of “How can the semantic interpretation of a formal symbol system be made intrinsic to the system” [1]. While the Chinese room thought experiment addresses precisely the question whether machines can understand semantic content, the frame problem is a more specific problem of how to limit the context of propositions, and in its formal aspect, it is considered fairly solved [2].
AUTHORS:We did not refer to the more formal aspect of the frame problem. Accordingly, now specify that we are referring to “generalizations” of the frame problem, and added a citation to the work of Wheeler to clarify which versions of the frame problem we have in mind here (see line 81, and also our response to the next point).
Suggestion: On lines 69 and 70, use a formulation more similar to lines 171 and 172: “a variety of practical and theoretical problems”
AUTHORS:We have now changed the sentence to the following: “Over the years this overarching problem of meaning has been famously discussed in terms of a variety of more specific practical and theoretical problems, including the symbol grounding problem [11], the Chinese room argument [12], and generalizations of the frame problem [13,14].” (lines 79-81).
2.2- Imprecise terminology about enactivism.
Not all enactive theories are non-representationalist. Only the radical views abandon the notion of representation.
Suggestion: revise use of terminology when it comes to 4E cognition. Make it clear what are the differences with the radical views.
AUTHORS:We disagree with the reviewer that only radical enactivism abandons the notion of representation. Ever since the Varela et al.’s (1991) The Embodied Minddifferent traditions of enactivism have stood out from the rest of cognitive science, even from much of embodied, embedded, and extended cognitive science (i.e. most of the research that also goes under the label of 4E cognition), because of its rejection of explanatory appeals to internal mental representations. We now clarify in the introduction that the key contrast that we have in mind is between representationalist and non-representationalist approaches: “Accordingly, there is a growing consensus that new approaches are required, especially in the form of non-representationalist embodied, dynamical, and enactive approaches” (lines 43-49).
2.3- Conflating the notion of agency with attribution of meaning.
The notion of agency developed by Barandiaran et, al. [3], does not claim that agency entails semantic understanding.
There is an important notion when it comes to Barandiaran at.al.’s definition of agency, that is neglected by the authors, namely, asymmetry. This notion explains the difference between the proto-cell example and a system that can be considered an agent.
Asymmetry, in my understanding, could also give room to indeterminacy, which would be a feature of the agent’s actions. But this is debatable.
Suggestion: consider the conceptual role of the notion of asymmetry.
AUTHORS:We agree that asymmetry is an interesting aspect of agency, but we are not sure that asymmetry by itself could give rise to indeterminacy. It seems to us that Barandiaran et al. assume that asymmetry is compatible with a concept of nature characterized by determinism and completeness, as suggested by their later dynamical systems models of agency.
2.4- How would you define subjectivity in Nature--?
The idea that the absence of objectivity in nature gives room for meaning is not straight-forward. If I understood it correctly, keeping the idea that meaning is given in subjectivity wouldn’t work after the dissolution of the subjective vs. objective duality.
Suggestion: address the question of how the absence of objectivity will leave room for meaning more directly.
AUTHORS:As we clarified in response to point 1, our aim is not to dissolve the difference between subjectivity and objectivity. The distinction between subjective and objective remains. This is why the objective has to make room for the subjective. We would like to propose a “hybrid” view of nature, in which the universe is not one-sidedly objective, but rather allows a coexistence of the objective and the subjective.
2.5- Specific Lines:
47 - What is meaning as such? Perhaps make reference to semantic content, maybe in a footnote?
AUTHORS:We prefer not to restrict the meaning of meaning to semantic content. There could also be non-contentful definitions of meaning in terms of relevance. We now clarify that we use the qualifier “as such” to mean “on its own terms” (line 53).
245 - What is a relative lack? Perhaps explain relative to what?
AUTHORS:For clarity we now added “compared to those” of an ordinary object (line 310).
316-320 - This explanation seems too short.
AUTHORS:We are not offering an explanation of mental causation, and neither is Jonas for that matter. We are giving a brief description of a situation in which a subjective intention could at least in principle make a difference in the physical world, even if we still do not fully understand how this effect is realized in practice. We now clarify our aim with by adding the following sentence at the end of the paragraph: “We are aware that this first-person recognition of the role of normativity in acting does not amount to an explanation of how a subjective intention can make a difference for objective movement in practice, but at least this recognition accepts that an intention could make a difference in principle.” (lines 424-427)
3- Corrections, optional suggestions, and discussion points:
(Optional)
3.1 Lines:
14 - replace ‘show up’ for ‘appear’
AUTHORS: Corrected.
46 - replace ‘the enactive approach’ for ‘radical enactive approaches’
AUTHORS:We prefer not to limit ourselves to the radical enactivist tradition promoted by Hutto, Myin, and others.
183 - correct: ‘a sub-symbolic architectures’
AUTHORS: Corrected.
191 - correct: representation ‘of’ the agent’s…
AUTHORS: Corrected.
217- In this sense (comma)
AUTHORS: Corrected.
233 - replace ‘mental impotence’ for a more precise expression
AUTHORS:We now added a subclause to define more precisely what we mean by mental impotence: “the incapacity of the subjective to make a difference over and above the objective” (lines 296-299).
241-246 - Give an example
AUTHORS:We have added the following paragraph: “The upshot is that while we can try to approximate living beings as state-determined systems, and even do so with some success, ultimately, their irreducible subjective nature prevents this partial approximation from being turned into a complete specification. Moreover, we argue that this is not a practical limitation due to measurement problems or a lack of knowledge. Rather, if our proposal is correct, this is a necessary limitation when approaching the subjective via an objective perspective, and as such the limitation can only ever be fully overcome at the price of subjectivity itself: A living being can never be fully known objectively as long as it is living, but only when it is dead and hence no longer behaves in accordance with subjective norms. Nevertheless, this in principle limitation of complete objective determination should itself be objectively detectable in practice. For example, if we could adapt the classical two-slit experiment of quantum physics to living beings, such that a living being has to pass between one of two exactly identical slits in a wall, we predict that, just like in the original particle version, it would be in principle impossible to predict the outcome on a trial-by-trial basis, no matter how simple the organism and no matter how much we know about the situation. This kind of experiment would then allow us to try to better understand the processes that give rise to this uncertainty at the behavioral level, i.e. those which make room for the subjective.” (lines 311-330).
273-275 - Give an example
AUTHORS:We have added the following paragraph: “To illustrate this possibility, it is helpful to remember that ethology and psychology have in practice already developed effective methods of working with the uncertainty of the behavior of the living, namely in terms of probability functions. The key theoretical difference, from the perspective of the nature-- strategy we are proposing, is that it is misguided to explain away all of this uncertainty by attributing it to practical limitations of current scientific practices. To the contrary, if we are correct in claiming that this uncertainty is in fact a result of the indeterminacy through which the subjective can express itself in objective terms, no future or even idealized scientific practice would be able to eradicate it completely. Accordingly, the nature-- strategy may provide a promising starting point from which to develop an alternative theory of agency and behavior that can make better sense of psychology’s widely publicized replication crisis.” (lines 361-370)
309-311 - Sentence is too long and confuse
AUTHORS: We divided this sentence into two parts.
313 - delete ‘for’ (I grasp my cup)
AUTHORS: Replaced with “I reach for my cup”. (line 416)
318 - replace ‘can I’ for ‘I can’
AUTHORS: Corrected.
318 - delete ‘my’ (from the first-person perspective)
AUTHORS: Corrected.
284- replace ‘And so it is’
AUTHORS: Replaced with “Accordingly, it is”. (line 385)
3.2 - It seems to me that if we consider nature as essentially non-deterministic and incomplete, the interaction between organism and environment ceases to be determined not due to the agent’s ‘freedom’ - which is a requirement for meaning - but due to the very nature of nature. How would this affect your notion of meaning?
AUTHORS: In our view, there is no dichotomy between nature and freedom. Rather, freedom is a part of nature. We have to at least understand nature as allowing freedom. Such a concept of nature is what we propose in this paper. A detailed discussion about the distinction between freedom and general indeterminacy of nature is beyond the scope of this paper. (It is a highly important question, but too big a problem to deal with here. We just discussed a necessary condition for freedom.)
3.3 - I had the impression that you first opt for dissolving the duality between subjectivity and objectivity (lines 243-245), and later on you use the notion of subjectivity to justify meaning from the first person perspective (317-320). If the aim is to replace the duality with something neutral between them (334), how would meaning be subjective?
AUTHORS: As we mentioned in response to point 1, we do not opt for dissolving the duality between subjectivity and objectivity . We are interested in exploring how they could be intertwined.
Another question is: How could I see meaning in the behavior of others?
AUTHORS: This is an interesting question, but not central to our current purposes. Here we are mainly interested in trying to understand how subjective meaning could affect objective behavior. Future work could try to explore what our position of the intertwinement of the subjective and the objective implies for phenomenological accounts of direct social perception.
3.4 - I can imagine an artificial agent who makes decisions, based on quantum models, which give room for indeterminacy, and still doesn’t grasp the meaning of its actions. I see that at the end of the paper you mention that indeterminacy would not be sufficient, but I’m not quite convinced that it would be necessary either. Perhaps there is a fine distinction on the notion of indeterminacy that we are not contemplating.
AUTHORS: The artificial agent you imagine exactly shows that indeterminacy is not sufficient for meaning. Working out the sufficient conditions is an interesting topic for future research. As for the necessity of indeterminacy, we would like to repeat our claim that there is no room for meaning to make a difference in a completely deterministic explanation of nature.
When would you say that a ‘natural’ agent (someone) understood a rule, or grasped the meaning?
AUTHORS:These are important questions, but to some extent our proposal is neutral with respect to what exactly constitutes a subject’s understanding of meaning. For our purposes it is sufficient to take meaning as a given and then ask how that subjective meaning could make a difference in an objective world. We thank the reviewer for helping us to better clarify the scope of our contribution. We now state this scope explicitly: “We will not enter more deeply into the enactive approach to grounding meaning here, because our primary concern lies with how meaning, once present, could make a difference for behavior in the physical world, no matter how meaning arises in the first place.” (lines 140-142)
I can also imagine an agent that is not free and yet understands its actions. (I cannot avoid acting the way I act, nevertheless I know why I act this way.)
AUTHORS: We would argue that if an agent really cannot avoid a bodily movement, then we cannot call it “acting” in a genuine sense. In that case, the agent does not act, with all the normativity that this would entail, but is just forced to move by objective causes. We do not deny that it is possible that our body can undergo movements that are forced upon it by objective causes and still understand why this is happening, like involuntarily being moved around by waves while swimming at the beach. What we are claiming is that if the only movements that would be possible in the world were such involuntary objectively determined movements and nothing else, then meaning could not make a difference to movement.
Is being free a necessary condition for understanding?
AUTHORS: We thank the reviewer for prompting us to think about this question. The autopoietic tradition of enactivism shares with Jonas the intuition that it is the precarious existence of the living, which makes the living both independent and yet dependent on the external world, and which Jonas characterizes as a “needful freedom,” that ultimately grounds the possibility of understanding things with a concerned or caring perspective. We have now clarified this relationship with our proposal in section 2 by adding the following paragraph: “But even in the case of real living beings, it only makes sense to claim that their meaningful perspective is grounded in their precarious existence if we accept, as Jonas [35] proposed, that their being is their own doing. An organism actively brings about its own physical existence, and it is this internal relationship between doing and being that makes the organism a being that is concerned with what it is doing, in particular with its continued self-preservation [36]. Of course, this is just an initial sketch and more has to be said about how the enactive approach could account for norms that are not related to the organism’s need to avoid dying [37,38], for instance by developing detailed accounts of additional processes of self-generation that are hierarchically decoupled from metabolism [23]. Nevertheless, a fundamental worry would remain: to what extent are we justified in claiming that the organism’s being is something actively done, rather than merely passively undergone, if all of its unfolding processes are completely prespecified by a deterministic universe? Such a universe is not compatible with Jonas’ characterization of life as “needful freedom” [20]. A precarious existence may be necessary for a meaningful perspective, but so is the freedom to make a genuine difference with respect to the needs of this existence.” (lines 126-139)
3.5- The main argument, namely, that a re-conception of the notion of nature should take place, is based on the idea that adaptive autopoietic systems are insufficient for meaning. If we reduce the argument into simple steps it would be like this:
We need to understand meaning in order to advance in AI
The conception of adaptive autopoietic system is not sufficient to account for meaning
Therefore we should reconsider our notion of nature
Of course, this reduction is neglecting several steps, but based on this impression that I had from the text, I would suggest making it clear that your suggestion of a re-conception of the notion of nature doesn’t follow from the fact that the concept of adaptive autopoietic systems is insufficient for meaning.
AUTHORS: This suggestion is helpful. We should have emphasized that a re-conception of the notion of nature does not followfrom the fact that the concept of adaptive autopoietic systems is insufficient. This is not the reason for, but just motivates a re-conception of nature. We now make this explicit during our discussion of the problems faced by the theory of adaptive autopoietic systems in section 2: “We believe that this impotence of subjectivity is too steep a price to pay for the naturalization of meaning, and we are therefore motivated to explore alternatives.” (lines 185-186)
4- Overall Recommendation
Accept - revisions required.
I suggest the authors consider the points mentioned in item 2. Suggestions on item 3 are optional to improve readability. Discussion points are also optional.
5- References
[1] Harnad, S. (1990). The symbol grounding problem. Physica D: Nonlinear Phenomena, 42(1), 335–346. https://doi.org/10.1016/0167-2789(90)90087-6
[2] Shanahan, M. (2016). The Frame Problem. In E. N. Zalta (Ed.), The Stanford Encyclopedia of Philosophy (Spring 2016). Metaphysics Research Lab, Stanford University. Retrieved from https://plato.stanford.edu/archives/spr2016/entries/frame-problem/
[3] Barandiaran, X. E., Di Paolo, E., & Rohde, M. (2009). Defining Agency: Individuality, Normativity, Asymmetry, and Spatio-temporality in Action. Adaptive Behavior, 17(5), 367–386. https://doi.org/10.1177/1059712309343819
AUTHORS: We thank the reviewer for all of these detailed observations. We have gone through the manuscript one more time to clarify and sharpen our arguments based on all of the points that were raised.